# DEEP ENSEMBLE KERNEL LEARNING

## ABSTRACT

Gaussian processes (GPs) are nonparametric Bayesian models that are both flexible and robust to overfitting. One of the main challenges of GP methods is selecting the kernel. In the deep kernel learning (DKL) paradigm, a deep neural network or "feature network" is used to map inputs into a latent feature space, where a GP with a "base kernel" acts; the resulting model is then trained in an end-to-end fashion. In this work, we introduce the "deep ensemble kernel learning" (DEKL) model, which is a special case of DKL. In DEKL, a linear base kernel is used, enabling exact optimization of the base kernel hyperparameters and a scalable inference method that does not require approximation by inducing points. We also represent the feature network as a concatenation of an ensemble of learner networks with a common architecture, allowing for easy model parallelism. We show that DEKL is able to approximate any kernel if the number of learners in the ensemble is arbitrarily large. Comparing the DEKL model to DKL and deep ensemble (DE) baselines on both synthetic and real-world regression tasks, we find that DEKL often outperforms both baselines in terms of predictive performance and that the DEKL learners tend to be more diverse (i.e., less correlated with one another) compared to the DE learners.

## 1 INTRODUCTION

In recent years, there has been a growing interest in Bayesian deep learning (DL), where the point predictions of traditional deep neural network (DNN) models are replaced with full predictive distributions using Bayes' Rule (Neal, 2012; Wilson, 2020). The advantages of Bayesian DL over traditional DL are numerous and include greater robustness to overfitting and better calibrated uncertainty quantification (Guo et al., 2017; Kendall & Gal, 2017). Furthermore, the success of traditional DL already rests on a number of probabilistic elements such as stochastic gradient descent (SGD), dropout, and weight initialization– all of which have been given Bayesian interpretations (Smith & Le, 2018; Gal & Ghahramani, 2016; Kingma et al., 2015; Schoenholz et al., 2016; Jacot et al., 2018), so that insights into Bayesian DL may help to advance DL as a whole.

Gaussian processes (GPs) are nonparametric Bayesian models with appealing properties, as they admit exact inference for regression and allow for a natural functional perspective suitable for predictive modeling (Rasmussen & Williams, 2005). While at first glance GPs appear unrelated to DL models, a number of interesting connections between GPs and DNNs exist in the literature, suggesting that GPs can constitute a valid approach to Bayesian DL (Neal, 1996; Lee et al., 2018; de Matthews et al., 2018; Jacot et al., 2018; Damianou & Lawrence, 2013; Salimbeni & Deisenroth, 2017; Agrawal et al., 2020). A GP prior is typically characterized by its covariance function or "kernel", which determines the class of functions that the GP can model, as well as its generalization properties outside training data. Kernel selection is the primary problem in GP modeling, and unfortunately traditional kernels such as the radial basis function (RBF) kernel are not sufficiently expressive for complex problems where more flexible models such as DNNs generally perform well. This is the key motivation for kernel learning, which refers to the selection of an optimal kernel out of a family of kernels in a data-driven way.

A number of approaches to kernel learning exist in the literature, including some that parameterize kernels using DNNs (Zhou et al., 2019; Li et al., 2019; Bullins et al., 2018; Sinha & Duchi, 2016). As these approaches involve learning feature representations, they are fundamentally different from random-feature methods for efficient kernel representation (Rahimi & Recht, 2007; 2008). However, these approaches are not specific to GPs and do not take advantage of a robust Bayesian framework.

In contrast, the deep kernel learning (DKL) paradigm does exactly this; In DKL, a DNN is used as a feature extractor that maps data inputs into a latent feature space, where GP inference with some "base kernel" is then performed (Wilson et al., 2016b;a; Jean et al., 2016; Al-Shedivat et al., 2017; Bradshaw et al., 2017; Izmailov et al., 2018; Xuan et al., 2018). The resulting model is then trained end-to-end using standard gradient-based optimization, usually in a variational framework. We note that the DKL model is just a GP with a highly flexible kernel parameterized by a DNN. By optimizing all hyperparameters (including the DNN weights) with type II maximum likelihood estimation, the DKL model is able to learn an optimal kernel in a manner directly informed by the data, while also taking advantage of the robustness granted by the Bayesian framework. A special case of DKL that is worthy of note was considered in Dasgupta et al. (2018), who use a linear base kernel and impose a soft orthogonality constraint to learn the eigenfunctions of a kernel. Although similar in spirit to the approach in this paper, their method does not make use of an efficient variational method, nor is distributed training made possible since all of the basis functions are derived from the same feature network.

In this work, we introduce the "deep ensemble kernel learning" (DEKL) model– a simpler and more efficient special case of DKL with two specifications– the base kernel is linear, and the feature network is partitioned into an "ensemble" of "learners" with common network architecture. In contrast to nonlinear kernels, the linear kernel allows us to derive an efficient training and inference method for DEKL that circumvents the inducing points approximation commonly used in traditional DKL. The hyperparameters of the linear kernel can also be optimized in closed form, allowing us to simplify the loss function considerably. Convenience aside, we show that DEKL remains highly expressive, proving that it is universal in the sense that it can approximate any continuous kernel so long as its feature network is arbitrarily wide. In other words, we may keep the base kernel simple if we are willing to let the feature network be more complex. The second specification of DEKL lets us handle the complexity of the feature network; because the feature network is partitioned, it admits easy model parallelism, where the learners in the ensemble are distributed. Moreover, our universality result only requires the number of learners to be arbitrarily large; the learners themselves need not grow (meaning fixed-capacity learners are sufficient), avoiding additional model parallelism.

From a different perspective, DEKL may be regarded as an extension of traditional ensembling methods for DNNs and in particular the deep ensemble (DE) model of Lakshminarayanan et al. (2017), which is also highly parallelizable. In a DE, each DNN learner parameterizes a distribution over the variates (e.g., the mean and variance of a Gaussian in regression, or the logits of a softmax vector in classification). Each learner is trained independently with maximum likelihood estimation, and the final predictive distribution of the DE is then defined to be a uniform mixture of the individual learner predictive distributions. Although not Bayesian itself, the DE model boasts impressive predictive performance and was shown to outperform Bayesian methods such as probabilistic back propagation (Hernández-Lobato & Adams, 2015) and MC-dropout (Gal & Ghahramani, 2016). In contrast, in DEKL, the learners are trained jointly via a shared linear GP layer. We surmise that this may help to promote diversity (i.e., low correlation) among the learners by facilitating coordination, which we verify experimentally. Unlike non-Bayesian joint ensemble training methods such as that of Webb et al. (2019), we hypothesize that the DEKL learners might learn to diversify in order to better approximate the posterior covariance– an inherently Bayesian feature. We therefore expect DEKL to be more efficient than DKL and more robust than DE, by drawing on the strengths of both (see Fig. 1 for a comparison of model architectures).

## 2 DEEP ENSEMBLE KERNEL LEARNING

A DKL model is a GP whose kernel encapsulates a DNN for feature extraction (Wilson et al., 2016b;a). A deep kernel is defined as

$$K_{\text{deep}}(x_1, x_2; \theta, \gamma) = K_{\text{base}}(\varphi(x_1; \theta), \varphi(x_2; \theta); \gamma), \tag{1}$$

where $\varphi(\cdot; \theta)$ is a DNN with weight parameters $\theta$ and $K_{\text{base}}(\cdot, \cdot; \gamma)$ is any chosen kernel—called the "base kernel"—with hyperparameters $\gamma$. Note that the kernel hyperparameters of $K_{\text{deep}}$ include all hyperparameters $\gamma$ of the base kernel $K_{\text{base}}$ as well as the DNN weight parameters $\theta$; Given the expressive power of DNNs, the deep kernel is also highly expressive and may be viewed as a method to automatically select a GP model.

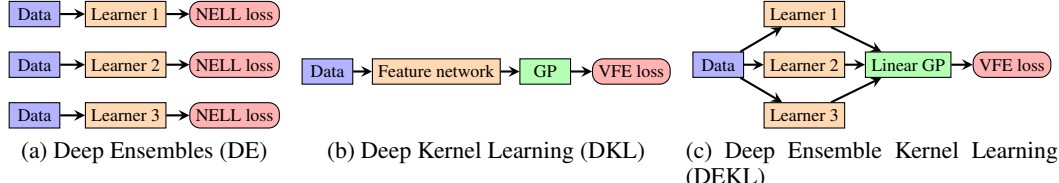

(a) Deep Ensembles (DE)    (b) Deep Kernel Learning (DKL)    (c) Deep Ensemble Kernel Learning (DEKL)

Figure 1: Training various neural architectures considered in this work. Deep ensembles (a) use identical prediction networks with independent initialization and training. Deep kernel learning (b) uses a neural feature extraction network with an expressive kernel such as an RBF. Our model, deep ensemble kernel learning (c) uses an ensemble of feature networks trained jointly through a linear kernel that enables exact inference via minimization of variational free energy for regression.

Our proposed method, DEKL, is a special case of DKL. Whereas RBF and Matern kernels are typically used as base kernels in DKL, in DEKL we take the base kernel to be the linear kernel:

$$K_{\text{lin}}(x_1, x_2; V) = x_1^\top V x_2, \tag{2}$$

where $V$ is a symmetric positive-semidefinite matrix. In order to enable parallel computation, in DEKL, we use a feature network $\varphi(\cdot; \theta)$ that is partitioned into an "ensemble" of subnetworks $\varphi_i(\cdot; \theta_i)$ called "learners", having identical network architectures; i.e., $\varphi(\cdot; \theta)$ is the concatenation of the outputs of the $\varphi_i(\cdot; \theta_i)$. DEKL offers a number of advantages over general DKL:

1. Unlike the hyperparameters of the RBF kernel, in DEKL, we can optimize the hyperparameters of the linear base kernel in closed form.

2. The linear base kernel allows us to think of a DEKL model not just as a GP but as a finite-dimensional Bayesian linear model (BLM), conditional on the feature networks; this lets us derive an efficient inference method that is much simpler than the inducing points method used in general DKL.

3. Finally, the partitioned architecture of the feature network makes it much more amenable to model parallelism.

Note that DEKL is fundamentally different from random-feature methods such as that of Rahimi & Recht (2008), where the learners $\varphi(\cdot; \theta_i)$ are random features that are not optimized during training.

A potential drawback to partitioning the feature network as we do in DEKL is that, compared to general DKL, the network is less expressive. However, the following universal approximation theorem for DKL implies that this effect can be compensated by adding parallel learners:

**Theorem 1** (Universal kernel approximation theorem). *Let $\mathcal{X} \subset \mathbb{R}^D$ be some compact Euclidean domain, and let $\sigma : \mathbb{R} \to \mathbb{R}$ be a non-polynomial activation function (Pinkus, 1999). Then, given a continuous, symmetric, positive-definite kernel $\kappa : \mathcal{X} \times \mathcal{X} \mapsto \mathbb{R}$ and any $\epsilon > 0$, there exist a finite number $H$ of affine functions $\beta_i : \mathcal{X} \xrightarrow{aff.} \mathbb{R}$ and a symmetric positive semi-definite matrix $V \in \mathbb{R}^{H \times H}$ such that for any $x_1, x_2 \in \mathcal{X}$,*

$$\left| \sum_{i,j=1}^{H} v_{ij} \sigma(\beta_i(x_1)) \sigma(\beta_j(x_2)) - \kappa(x_1, x_2) \right| < \epsilon. \tag{3}$$

The proof, found in Appendix A, contains a straightforward combination of Mercer's Theorem (Mercer, 1909) and the Universal Approximation Theorem, attributed to Cybenko, Hornik, Leshno, and Pinkus (Cybenko, 1989; Hornik et al., 1989; Leshno et al., 1993; Pinkus, 1999). Note that Thm. 1 lets us represent non-stationary continuous kernels, in contrast to methods such as random Fourier feature expansion (Rahimi & Recht, 2007).

The approximation in Thm. 1 requires a possibly large number $H$ of affine functions $\beta_i$. However, in DEKL we replace the functions $x \to \sigma(\beta_i(x))$ with an ensemble of strictly more flexible DNN learners $\varphi(\cdot; \theta_i)$, which can help to reduce the number $H$ of learners required; in the proof of Thm. 1,

we approximate each eigenfunction of the target kernel $\kappa$ with a linear combination of the learners; if the learners are sufficiently expressive, then it may take only one learner per eigenfunction to approximate the target kernel. We also allow each learner $\varphi(\cdot; \theta_i)$ to have multiple outputs $M$. In the case of the simple learners $x \to \sigma(\beta_i(x))$, a learner with $M$ outputs is simply a concatenation of $M$ single-output learners, suggesting that multi-output learners may help to further reduce the number $H$ of required learners.

The summation in Eq. 3 should be understood as a deep kernel as in Eq. 1, where the DNN $\varphi$ in Eq. 1 is the concatenation of all learners and the base kernel is given by the linear kernel in Eq. 2. Theorem 1 is thus a statement about the universality of DEKL with a linear base kernel. Given that other kernels such as the RBF are more popular choices of base kernel in the DKL literature, it is natural to wonder if DEKL remains a universal kernel approximator if we change the base kernel. It turns out that not all choices of base kernel give universality, as is implied by the following remark.

**Remark 2.** *For a deep kernel (Eq. 1) with base kernel $K_{base} : \mathbb{R}^H \times \mathbb{R}^H \mapsto \mathbb{R}$ to be a universal kernel approximator, the base kernel must be unbounded both above and below.*

We give more details in Appendix B, but intuitively, since the base kernel is the outermost function in the deep kernel, any bound on its range will prevent the deep kernel from approximating kernels with unbounded range, such as the dot product kernel. The class of base kernels with bounded (or half-bounded) range, and thus the base kernels that do not give universality, is large and includes many popular kernels such as the RBF kernel and periodic kernel. The linear base kernel in Eq. 2 is therefore special, as it is not only convenient but also grants us universality.

We note the converse of Remark 2 is not true; an unbounded base kernel does not guarantee that a DEKL model is a universal kernel approximator. For example, restricting the matrix $V$ in the linear base kernel (Eq. 2) to a diagonal matrix breaks universality; this is because Thm. 1 must hold for even the simplest learners $x \to \sigma(\beta_i(x))$, which fails to happen when we restrict $V$ (see Appendix B for details). Classifying all base kernels for which DEKL is universal remains an important open problem for future work.

## 2.1 Exact Inference

Consider a dataset $\{(x_i, y_i) \in \mathbb{R}^D \times \mathbb{R}\}_{i=1}^N$. In DEKL, we model this data with likelihood $p(y \mid f)$ and GP prior $\mathcal{GP}(0, K_{\text{DE}}(\cdot, \cdot; \theta, V))$, where the kernel $K_{\text{DE}}(\cdot, \cdot; \theta, V) : \mathbb{R}^D \times \mathbb{R}^D \mapsto \mathbb{R}$ is defined as

$$K_{\text{DE}}(x_1, x_2; \theta, V) = \Phi(x_1; \theta)^\top V \Phi(x_2; \theta), \tag{4}$$

where $V$ is an $H \times H$ symmetric positive semidefinite matrix and $\Phi(x; \theta) = \{\varphi(x; \theta_i)\}_{i=1}^H$ is the concatenation of $H$ DNN learners $\varphi(\cdot; \theta_i) : \mathbb{R}^D \mapsto \mathbb{R}$ with common network architecture. The DEKL model is thus a standard GP but with a special kernel, and thus one may proceed following standard practices for GP inference. However, in contrast to other popular GP kernels such as the RBF kernel, the kernel in Eq. 4 has rank at most $H$, and we can leverage this property to derive a more efficient GP inference method by regarding it as a Bayesian linear model (BLM) where the learners act as the BLM basis functions. More precisely, our GP prior is equivalent to taking the latent function $f : \mathbb{R}^D \mapsto \mathbb{R}$ to be a generative model of the form

$$f(x) = \Phi(x; \theta)^\top a, \quad a \sim \mathcal{N}(0, V).$$

Now formulated as a BLM, we may perform Bayesian inference by inferring the posterior on $a$; when the number of training points $N$ is large, this approach is significantly more efficient than direct GP inference, which requires a Cholesky decomposition of complexity $O(N^3)$.

In the case of regression with a Gaussian likelihood, the posterior on $a$ admits a closed form. Moreover, the evidence of the above model given the training data also admits a closed form, allowing us to obtain maximum evidence estimates of the DEKL hyperparameters $V$ and $\theta$ as is standard practice in GP methods. However, the dependence of the evidence on $\theta$ is in general complex, so that the maximum evidence estimates must be approximated. Moreover, the evidence is not separable across the training data, so that optimization is not amenable to SGD with minibatching and is thus not scalable to large datasets. We therefore perform approximate inference and maximum evidence estimation in a variational framework, discussed in the next section. We reiterate that DEKL admits exact inference on small datasets and that maximum evidence estimation can be performed through non-stochastic gradient-based optimization methods; however, we consider the variational framework as it can handle a larger class of problems.

## 2.2 Variational inference

In variational inference, we approximate the posterior distribution on the vector $a$ with a variational distribution by solving an optimization problem. In the GP literature, it is standard to take the variational distribution to itself be a GP. In our DEKL model formulated as a BLM, this corresponds to a normal variational distribution $\mathcal{N}(\mu, \Sigma)$ on $a$, where $\mu \in \mathbb{R}^H$ and $\Sigma \in \mathbb{R}^{H \times H}$. We now minimize the KL divergence between the variational and posterior distributions on $a$, which is equivalent to minimizing the variational free energy loss function:

$$\mathcal{L}_{\text{VFE}}(\mu, \Sigma, \theta, V) = \sum_{i=1}^{N} \text{NELL}_i(\mu, \Sigma, \theta) + \text{KL}(\mu, \Sigma, V) \tag{5}$$

$$\text{NELL}_i(\mu, \Sigma, \theta) = -\mathbb{E}_{a_i \sim \mathcal{N}(\mu, \Sigma)} \left[ \log p(y_i \mid \Phi(x_i; \theta)^\top a_i) \right] \tag{6}$$

$$\text{KL}(\mu, \Sigma, V) = \text{D}_{\text{KL}}(\mathcal{N}(\mu, \Sigma) \mid \mathcal{N}(0, V))$$

$$= \frac{1}{2} \mu^\top V^{-1} \mu + \frac{1}{2} \text{tr}(\Sigma V^{-1}) - \frac{1}{2} \log \det(\Sigma V^{-1}) - \frac{H}{2}. \tag{7}$$

Observe that we optimize both the variational parameters $\mu$ and $\Sigma$ as well as the kernel hyperparameters $\theta$ and $V$ as is standard in GP inference. Note also that this loss function is separable over the training data and is thus amenable to optimization with SGD and minibatching; In this paper, we focus on regression tasks with Gaussian likelihood $p(y \mid f) = \mathcal{N}(y; f, \tau^{-1})$ with precision $\tau$. The corresponding NELL admits a closed form and is given by

$$\text{NELL}_i(\mu, \Sigma, \theta) = \frac{\tau}{2} \|\Phi(x_i; \theta)^\top \mu - y_i\|^2 + \frac{\tau}{2} \Phi(x_i; \theta)^\top \Sigma \Phi(x_i; \theta) - \frac{1}{2} \log \tau + \frac{1}{2} \log(2\pi). \tag{8}$$

Note that in practice, we parameterize $\Sigma$ with its positive-definite lower-triangular Cholesky factor in order to guarantee the symmetric positive-definiteness of $\Sigma$.

For scalability to large training sets, DKL inference is usually performed using the inducing points approximation in a sparse variational framework (Titsias, 2009; Hensman et al., 2013). In the case of DEKL, however, due to using a linear base kernel, the inducing points approximation is not necessary, as seen above. Instead, our parameters $\mu$ and $\Sigma$ take the place of the inducing parameters in the sparse variational GP framework, and we are able to maintain an exact model for the posterior GP.

## 2.3 Optimal prior covariance

The variational free energy depends on the prior covariance $V$ only through the KL term, which has a tractable form. This allows us to optimize $V$ in closed form; we find the optimal prior covariance to be

$$V_*(\mu, \Sigma) = \Sigma + \mu\mu^\top. \tag{9}$$

Next, we substitute this optimal prior covariance back into the KL term, thereby eliminating the prior covariance from the variational free energy altogether. After some simplification, the KL term takes the following form:

$$\text{KL}_O(\mu, \Sigma) = \text{KL}(\mu, \Sigma, V_*(\mu, \Sigma)) = \frac{1}{2} \log(1 + \mu^\top \Sigma^{-1} \mu). \tag{10}$$

(See Appendix C for derivations of Eqs. 9-10). During training, this term drives down the signal-to-noise ratio (SNR) $\mu\Sigma^{-1}\mu$, encouraging the model to learn a more robust fit to the data.

We contrast the form of Eq. 10 with the KL term obtained by fixing the prior covariance to the identity $V = I$:

$$\text{KL}_I(\mu, \Sigma) = \text{KL}(\mu, \Sigma, I) = \frac{1}{2} \|\mu\|^2 + \frac{1}{2} \text{tr}(\Sigma) - \frac{1}{2} \log \det(\Sigma) - \frac{H}{2}. \tag{11}$$

We note that while Eq. 11 includes a quadratic penalty on $\mu$, Eq. 10 includes only a logarithmic penalty; optimizing the prior covariance therefore results in weaker regularization of $\mu$. The same is true with respect to $\Sigma$; both Eqs. 10-11 include roughly logarithmic penalties on $\Sigma^{-1}$ that encourage larger $\Sigma$, but Eq. 11 additionally includes a linear penalty on $\Sigma$, resulting in stronger regularization of $\Sigma$. We note that in regression, the NELL term already favors small $\Sigma$ (see Eq. 8), so that the KL term is the only thing preventing $\Sigma$ from degenerating. By weakening the regularization on $\Sigma$, we suspect the KL term with optimal prior covariance to be beneficial in practice.

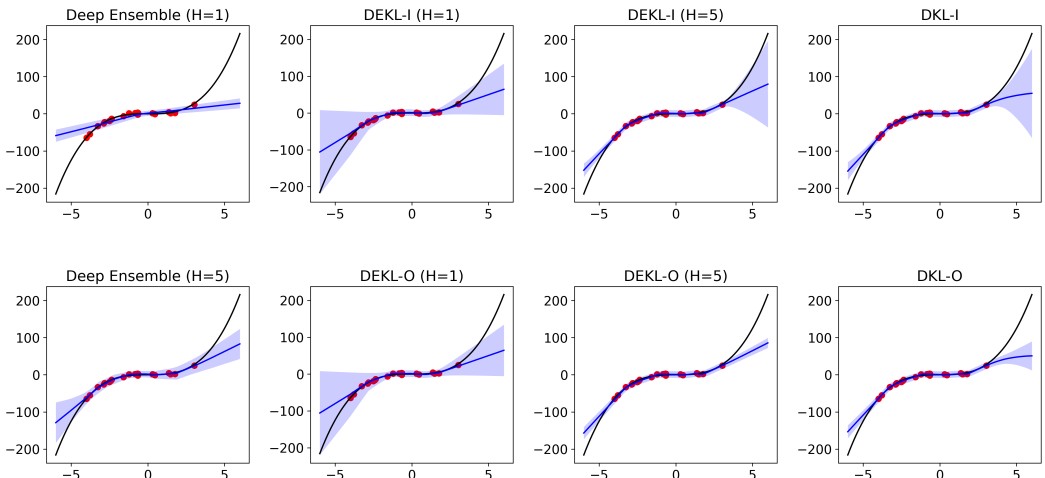

Figure 2: Synthetic cubic dataset. The first column shows a single DE learner and a DE model of five learners. The second and third columns show one DEKL learner and a DEKL model of five learners, both with and without optimal prior covariance. The fourth column shows a DKL model of the same width as the five-learner DEKL, both with and without optimal prior covariance. The predictive mean function (blue) with three units of standard deviation is shown for each model trained on the cubic dataset (red), with ground truth (black). Although the five-learner DEKL has far fewer parameters than DKL due to partitioning of the feature network, it approximates the full DKL well on this dataset, and improves over the equivalent DE model.

## 3 EXPERIMENTS

In our experiments, we evaluate DEKL on various regression datasets, comparing it to two key baselines– DE and DKL (with linear base kernel). For both DKL and DEKL, we also compare the effect of the KL term with optimal prior covariance, $\mathrm{KL}_O$ (see Eq. 10), to that of an identity prior covariance, $\mathrm{KL}_I$ (see Eq. 11).

### 3.1 SYNTHETIC DATA

For our first experiment, we consider a simple regression problem with one-dimensional inputs and outputs, as it allows for easy visualization of the posterior predictive distribution. For this purpose, we generate data via $y = x^3 + \epsilon$ where $\epsilon \sim \mathcal{N}(0, 3^2)$ as in Lakshminarayanan et al. (2017). We sample twenty $(x, y)$ pairs with $x \sim \mathrm{Unif}(-4, 4)$. As already stated, we compare the DEKL model to the DKL and DE baselines. For DE, we follow the implementation of Lakshminarayanan et al. (2017), except that we do not use adversarial samples, which the authors say is optional. For the DEKL and DE models, we use either one learner ($H = 1$) or five learners ($H = 5$), where each learner is a shallow MLP with rectified linear unit (ReLU) activation and 50 hidden neurons. For the DKL model, we use a shallow MLP feature network with ReLU activation and 250 hidden and five output neurons, so that it has the same total hidden width as the five-learner DE and DEKL models. Note that the only difference between the file-learner DEKL and DKL models is that—compared to the DKL model—the DEKL model is missing some connections in its second layer.

We train all models on the training set using the Adam optimizer with learning rate $10^{-3}$ and full batches for 1000 epochs. For the DEKL and DKL models, we set the noise precision to its true value $\tau = \frac{1}{9}$.

The predictive distributions of all models on their input domain are visualized in Fig. 2. We make a number of observations: First, the DKL and DEKL models with fixed prior display better uncertainty quantification than DE, having lower variance within the training domain (interpolation) and higher variance outside the training domain (extrapolation). Second, DE and DEKL with one learner have

less flexible means compared to their five-learner counterparts, which we expect due to reduced expressivity. However, DE is more confident with one learner while DEKL is less confident. This highlights the difference in the uncertainty quantification mechanisms of the two models; while DE relies on multiple learners for its predictive variance, DEKL offers some predictive uncertainty even with one learner thanks to Bayesian inference. Third, using the optimal prior covariance results in overconfidence. We suspect this is because theKL term in Eq. 10 imposes weaker regularization on $\mu$ than does the KL term in Eq. 11, which may be critical on a dataset of only 20 training points. Finally, the five-learner DEKL predictive distribution is very similar to that of DKL, indicating that the partitioning of the feature network in DEKL grants efficiency without hurting predictive performance.

## 3.2 REAL-WORLD DATA

We test our DEKL model on a set of regression tasks from the UCI repository that has become a popular benchmark in the Bayesian DL literature, starting with the work of Hernández-Lobato & Adams (2015). We again compare DEKL to the DE and DKL baselines as in Sec. 3.1, and we use the same experimental setup as in Lakshminarayanan et al. (2017). We only consider DE and DEKL with five learners, with each learner having 50 hidden neurons, except on the Protein dataset, where we use 100 hidden neurons (and correspondingly, 500 hidden neurons in the DKL models).

We consider 20 random train-test splits of each dataset except Protein, where we use only 5 train-test splits. For each split, we use $20\%$ of the training set for validation. We train all models on the training sets using the Adam optimizer with learning rate $10^{-3}$ and minibatch size 128 for 1000 epochs. For the DEKL and DKL models, we set the noise precision $\tau$ by performing a grid search and selecting the value that results in the lowest negative log-likelihood (NLL) on the validation set. For all models, we find the epoch where NLL is minimized on the validation set, then retrain all models on the full dataset up to that number of epochs, and finally evaluate the models on the test sets.

We report the test root mean square error (RMSE) and test NLL averaged over all train-test splits (Tables 1-2; for brevity, we list Table 2 in Appendix D, where we also include the definitions of RMSE and NLL). We see that DEKL often outperforms DE, especially in terms of RMSE, suggesting that joint ensemble Bayesian training can often be beneficial. A further comparison of DE and DEKL may be found in Appendix E, where we examine predictive performance as a function of number of learners and find that the performance gap between DE and DEKL is largely independent of the number of learners, at least up to five learners. We note that DEKL also attains either comparable or superior performance to other GP methods, both shallow and deep, with RBF kernel in terms of NLL (GP scores are listed in Salimbeni & Deisenroth (2017), who use the same experimental setup as we do). More surprisingly, DEKL often outperforms DKL, indicating that the benefit of DEKL over DKL goes beyond simple efficiency or ease of parallelization; partitioning the feature network can often lead to a significant boost in predictive performance as well. We also observe that $\text{KL}_I$ vs. $\text{KL}_O$ has little impact on performance, suggesting that either the data is rich enough that the non-diagonal terms of the prior covariance matrix have an insignificant effect on the model evidence, or that the feature network is expressive enough to include the Cholesky factor of the prior covariance matrix in the case of $\text{KL}_I$.

## 3.3 DIVERSITY OF LEARNERS

To see if joint ensemble training in a Bayesian framework promotes diversity among learners, we compare the diversity of learners in DEKL to that in DE on the UCI datasets. Given an ensemble of learners, we measure its diversity in terms of the functional correlation matrix of its learners, and we say that the ensemble is more diverse if the learners have lower correlation. We base this notion of diversity on the work of Brown et al. (2005), who showed that extending the classic bias-variance decomposition of the mean squared error to an ensemble of learners requires an additional term in the decomposition that quantifies the average covariance between distinct learners.

For each UCI dataset and each train-test split, we load the DE and DEKL models (with and without optimal prior covariance) already trained following Sec. 3.2 and evaluate their feature matrices on the entire dataset (including both training and test points). Each row of a feature matrix corresponds to an observation, and each column corresponds to the output of one of five learners. For DE, we

Table 1: Test-RMSE (mean and std. dev. across 20 train-test splits) on UCI datasets, using deep ensembles (DE), deep kernel learning (DKL) and deep ensemble kernel learning (DEKL). Suffices -I and -O indicate using $\mathrm{KL}_I$ (Eq. 11) and $\mathrm{KL}_O$ (Eq. 10), respectively. The numbers $N$ and $D$ are the size and dimension of each dataset respectively.

| Dataset | $N$ | $D$ | DE | DKL-I | DKL-O | DEKL-I | DEKL-O |
|---|---|---|---|---|---|---|---|
| Boston housing | 506 | 13 | $3.29 \pm 1.01$ | $3.13 \pm 0.77$ | $3.43 \pm 1.68$ | $3.03 \pm 0.87$ | $3.03 \pm 0.86$ |
| Concrete | 1030 | 8 | $5.79 \pm 0.82$ | $4.93 \pm 0.80$ | $4.79 \pm 0.67$ | $4.52 \pm 0.59$ | $4.57 \pm 0.64$ |
| Energy | 768 | 8 | $2.04 \pm 0.31$ | $0.61 \pm 0.22$ | $0.94 \pm 0.96$ | $0.48 \pm 0.06$ | $0.47 \pm 0.05$ |
| Kin8nm | 8192 | 8 | $0.08 \pm 0.00$ | $0.08 \pm 0.00$ | $0.08 \pm 0.01$ | $0.07 \pm 0.00$ | $0.07 \pm 0.00$ |
| Naval Propulsion | 11934 | 26 | $0.00 \pm 0.00$ | $0.00 \pm 0.00$ | $0.00 \pm 0.00$ | $0.00 \pm 0.00$ | $0.00 \pm 0.00$ |
| Power plant | 9568 | 4 | $3.95 \pm 0.16$ | $3.83 \pm 0.25$ | $3.85 \pm 0.23$ | $3.91 \pm 0.20$ | $3.91 \pm 0.18$ |
| Protein | 45730 | 9 | $4.40 \pm 0.12$ | $4.06 \pm 0.10$ | $3.94 \pm 0.07$ | $3.99 \pm 0.11$ | $3.95 \pm 0.02$ |
| Wine | 1599 | 22 | $0.63 \pm 0.03$ | $0.68 \pm 0.14$ | $0.67 \pm 0.06$ | $0.63 \pm 0.05$ | $0.64 \pm 0.05$ |
| Yacht | 308 | 7 | $0.72 \pm 0.24$ | $2.92 \pm 3.06$ | $2.52 \pm 2.78$ | $0.61 \pm 0.22$ | $0.62 \pm 0.22$ |

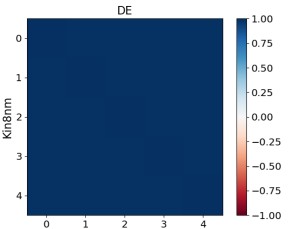 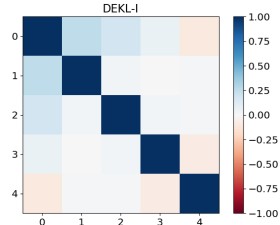 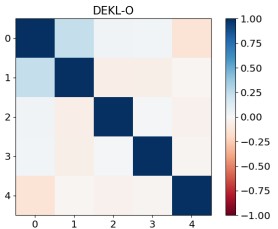

Figure 3: Correlation matrices of learners averaged over all train-test splits on the Kin8nm dataset using deep ensembles (DE) and deep ensemble kernel learning (DEKL). Suffices -I and -O indicate using $\mathrm{KL}_I$ (Eq. 11) and $\mathrm{KL}_O$ (Eq. 10), respectively.

use the mean output of each learner as one of the features. For DEKL, we multiply each column of the feature matrix ($\Phi(X; \theta)$ in the notation of Sec. 2.2 with $X$ the dataset) by the corresponding element of the parameter mean $\mu$ of the final BLM layer, and we additionally scale all columns by the number of learners (five); doing this, the DE and DEKL feature matrices are comparable in the sense that the predictive means of both models can be obtained as the average of the feature matrix columns. Finally, for each dataset, train-test split, and model, we calculate the correlation matrix of the feature matrix and then average all correlation matrices across the train-test splits.

We visualize the correlation matrices in Fig. 3. We show only one dataset here for brevity; results for all datasets may be found in Appendix F. Overall, DEKL results in lower correlation between learners and thus greater diversity, as hypothesized, and using the optimal prior covariance often boosts diversity further. However, more diversity does not necessarily imply better predictive performance; elucidating the relationship between diversity and predictive performance is a topic we leave for future work. We also observe extreme cases where the learners in DE are strongly correlated, while the learners in DEKL are almost entirely uncorrelated or orthogonal. This ability for the learners to approximately orthogonalize even in DEKL-I suggests that the individual learners are quite flexible and that the restriction of the prior covariance $V$ to the identity matrix is not a significant hindrance to expressive power; we suspect this may be why $\mathrm{KL}_I$ vs. $\mathrm{KL}_O$ has little impact on performance on the UCI datasets (Table 1).

## 4 CONCLUSION

We have introduced the DEKL model: a special case of DKL that is more easily parallelizable and thus more efficient than other DKL models of comparable network complexity. We have seen that DEKL often achieves better predictive performance than deep ensembles, which are also highly

parallelizable. Even with a linear base kernel, we have shown that DEKL is a universal kernel approximator if the feature network is allowed to be arbitrarily wide, and we handle the complexity of the feature network by partitioning it into an ensemble of learners. In our experiments, we found that DEKL often outperforms DE and promotes more diversity, suggesting that jointly training learners in a Bayesian framework can be beneficial. Interestingly, we also observed that DEKL often outperforms DKL of the same width, suggesting that partitioning the feature network not only grants computational efficiency through model parallelism but may also boost performance by reducing the parameter count compared to a fully-connected DKL.

We are considering several avenues for future work. We plan to study the scalability of DEKL to a very large number of learners in a distributed setting, which may be necessary to approximate a kernel of very high or infinite rank. A key challenge in this endeavor is that the size of the variational covariance $\Sigma$ grows quadratically with the number of learners; to deal with this, we will consider asymptotic regimes where efficient approximations of $\Sigma$ are justified, namely the many-simple-learners regime and the complex-learners regime. In the first of these regimes, the dimension $R$ of the span of $H$ learners is much less than $H$, which allows us to assume that $\Sigma$ is of low rank, specifically $R$. In the second regime, the learners may be sufficiently flexible to be orthogonal to one another, allowing us to assume $\Sigma$ is diagonal. This could lead to an approach similar to that of Dasgupta et al. (2018), but with distributed computation and using our variational framework. More generally, we may consider to approximate $\Sigma$ as a sum of a low-rank matrix and a diagonal matrix, possibly striking a balance between the two asymptotic regimes.

Another interesting topic of investigation is the mechanism by which DEKL achieves performance often superior to that of DE. In DE, the learners are trained independently, and thus any diversity among the learners is solely due to random initialization and the nonconvexity of the loss function. In contrast, in DEKL, the learners are able to "communicate" with one another through the common final GP layer, and thus we hypothesize that the learners may "coordinate" with one another to ensure diversity. Indeed, in our experiments, we found that DEKL does lead to greater diversity, but the precise mechanism by which this diversity emerges and how it impacts predictive performance are less clear. We believe that diversity in DEKL may be linked to the posterior covariance, which is generally nonzero in this Bayesian setting.

It is also tempting to extend DEKL to deep Gaussian processes (DGPs), which are models defined as compositions of GPs (Damianou & Lawrence, 2013; Cutajar et al., 2017; Salimbeni & Deisenroth, 2017). This will result in a DGP inference method that again does not require an inducing points approximation, similar to the random features expansion approach of Cutajar et al. (2017), except that the features are trainable DNNs that need not be orthogonal. Given that DEKL is a universal kernel approximator, we conjecture that a DGP with DEKL kernels can approximate any DGP with continuous kernels, under some distance metric on stochastic processes. Such a result would suggest a universal model for stochastic processes (or some suitably well-behaved subspace of them), arguably the most general universality theorem we can imagine in predictive modeling.

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

## A  PROOF OF UNIVERSAL KERNEL APPROXIMATION THEOREM

*Proof of Thm. 1.* By Mercer's Theorem (Mercer, 1909, Sec. 29), there exist non-negative scalars $\{\lambda_r\}_{r=1}^{\infty}$ and an orthonormal basis of continuous functions $\{e_r : \mathcal{X} \mapsto \mathbb{R}\}_{r=1}^{\infty}$ called "feature maps" such that the target kernel $\kappa$ admits the representation

$$\kappa(x_1, x_2) = \sum_{r=1}^{\infty} \lambda_r e_r(x_1) e_r(x_2), \tag{12}$$

where the sum on the righthand side converges absolutely and uniformly on the compact set $\mathcal{X}$. Selecting $0 < \delta < \epsilon$, uniform convergence guarantees the existence of an integer $R$ such that for all $x_1, x_2 \in \mathcal{X}$,

$$\left| \sum_{r=1}^{R} \lambda_r e_r(x_1) e_r(x_2) - \kappa(x_1, x_2) \right| < \delta. \tag{13}$$

By continuity of the feature maps $e_r$ and compactness of $X$, the feature maps are bounded, so there exists a finite positive $B > \max_{r=1,\ldots,R} \sqrt{\lambda_r} \sup_{x \in X} |e_r(x)|$. By the classic Universal Approximation Theorem (Pinkus, 1999), there exists a shallow MLP $\psi : \mathcal{X} \mapsto \mathbb{R}^R$ of width $H$ and parameters $u_{ir} \in \mathbb{R}$ and $\beta_i : \mathcal{X} \mapsto \mathbb{R}$ for $i = 1, \ldots, H$ such that for all $x \in \mathcal{X}$,

$$\psi_r(x) = \sum_{i=1}^{H} u_{ir} \sigma(\beta_i(x)) \tag{14}$$

$$\left| \psi_r(x) - \sqrt{\lambda_r} e_r(x) \right| < \gamma < \min\left( B, \frac{\epsilon - \delta}{3NB} \right) \tag{15}$$

$$|\psi_r(x_1)\psi_r(x_2) - \lambda_r e_r(x_1) e_r(x_2)| < \gamma(\gamma + 2B) < 3B\gamma \tag{16}$$

$$\left| \sum_{r=1}^{R} \psi_r(x_1)\psi_r(x_2) - \sum_{r=1}^{R} \lambda_r e_r(x_1) e_r(x_2) \right| < 3NB\gamma < \epsilon - \delta \tag{17}$$

$$\left| \sum_{r=1}^{R} \psi_r(x_1)\psi_r(x_2) - \kappa(x_1, x_2) \right| < (\epsilon - \delta) + \delta = \epsilon. \tag{18}$$

More explicitly, we have the uniform bound

$$\left| \sum_{i,j=1}^{H} \sum_{r=1}^{R} u_{ir} u_{jr} \sigma(\beta_i(x_1)) \sigma(\beta_j(x_2)) - \kappa(x_1, x_2) \right| < \epsilon. \tag{19}$$

Letting $v_{ij} = \sum_{r=1}^{R} u_{ir} u_{jr}$, the bound takes the form claimed in the statement of the theorem. Since this matrix $V$ is of the form $UU^T$, it is clearly symmetric and positive semi-definite.  □

## B  NON-UNIVERSAL DEKL

The kernel in Eq. 4 may be written as

$$K_{\text{DE}}(x_1, x_2; \theta, V) = K_{\text{lin}}(\Phi(x_1; \theta), \Phi(x_2; \theta); V), \tag{20}$$

where the linear base kernel $K_{\text{lin}}$ is given by Eq. 2. Theorem 1 then states that a kernel of this form can approximate any continuous target kernel to arbitrary accuracy, given some $\theta$ and $V$ (and sufficiently many learners); i.e., DEKL with a linear base kernel is universal. However, universality fails to hold if we replace the linear base kernel with certain other popular kernels.

First suppose we replace the linear base kernel in Eq. 20 with a base kernel $K_{\text{base}} : \mathbb{R}^H \times \mathbb{R}^H \mapsto \mathbb{R}$ whose range is bounded either above or below by $M$. Suppose we wish to use Eq. 20 to approximate the dot product kernel

$$K_{\text{dot}}(x_1, x_2) = a x_1^{\top} x_2, \quad a > 0,$$

where $x_1, x_2$ are taken to lie in the closed unit ball. This dot product kernel then has range $[-a, a]$. If we select the coefficient $a$ of the dot product kernel to be sufficiently large so that either $M < a - \epsilon$

(in case $K_{\text{base}}$ is bounded above) or $M > a + \epsilon$ (in case $K_{\text{base}}$ is bounded below) for some $\epsilon > 0$, then the deep kernel is incapable of approximating the constructed target dot product kernel, regardless of the feature network or learners used.

Next consider the linear kernel in Eq. 2 but where $V$ is restricted to a diagonal positive semidefinite matrix, and suppose we replace $K_{\text{lin}}$ in Eq. 20 with this new isotropic linear kernel. In case of arbitrarily complex learners, the resulting DEKL model is of course universal, as it is able to represent any target kernel as an arbitrary continuous feature embedding (thanks to the complexity of the learners) followed by an inner product. However, universality fails to hold if we take the learners to be simple. Consider learners of the form $\varphi(x; \beta_i) = \sigma(\beta_i(x))$ where $\beta_i$ is an affine map and $\sigma$ is the ReLU activation function. Since the diagonal elements of $V$ are non-negative to guarantee positive semidefiniteness, then the resulting DEKL kernel $K_{\text{DE}}$ is a sum-product of ReLU-activated functions with non-negative coefficients and is therefore itself non-negative. This kernel is therefore incapable of approximating target kernels that can return negative values, such as the dot product kernel.

## C   DERIVATION OF THE KL TERM WITH OPTIMAL PRIOR COVARIANCE

Here we derive the optimal prior covariance $V_*$ with respect to the variational free energy in Eq. 5 and use it to eliminate the prior covariance $V$ from the loss function entirely. We note that $V$ appears only in the KL term (Eq. 7) and only through its inverse $V^{-1}$. Minimizing the KL term with respect to $V^{-1}$, we obtain Eq. 9 as follows:

$$\frac{\partial}{\partial V^{-1}} \text{KL}(\mu, \Sigma, V) \mid_{V=V_*} = 0$$
$$\mu\mu^\top + \Sigma - (V_*^{-1})^{-1} = 0$$
$$V_* = \mu\mu^\top + \Sigma.$$

Substituting this back into Eq. 7 and applying the Matrix Inversion Lemma, we obtain Eq. 10 as follows:

$$
\begin{aligned}
\mathrm{KL}_O(\mu, \Sigma) &= \mathrm{KL}(\mu, \Sigma, V_*) \\
&= \frac{1}{2}\mu^\top(\mu\mu^\top + \Sigma)^{-1}\mu + \frac{1}{2}\mathrm{tr}[(\mu\mu^\top + \Sigma)^{-1}\Sigma] - \frac{1}{2}\log\det[(\mu\mu^\top + \Sigma)^{-1}\Sigma] - \frac{H}{2} \\
&= \frac{1}{2}\mu^\top\left(\Sigma^{-1} - \frac{\Sigma^{-1}\mu\mu^\top\Sigma^{-1}}{1 + \mu^\top\Sigma^{-1}\mu}\right)\mu \\
&\quad + \frac{1}{2}\mathrm{tr}\left[\left(\Sigma^{-1} - \frac{\Sigma^{-1}\mu\mu^\top\Sigma^{-1}}{1 + \mu^\top\Sigma^{-1}\mu}\right)\Sigma\right] \\
&\quad + \frac{1}{2}\log\left[\frac{\det(\mu\mu^\top + \Sigma)}{\det\Sigma}\right] - \frac{H}{2} \\
&= \frac{1}{2}\mu^\top\Sigma^{-1}\mu - \frac{1}{2}\frac{\mu^\top\Sigma^{-1}\mu\mu^\top\Sigma^{-1}\mu}{1 + \mu^\top\Sigma^{-1}\mu} \\
&\quad + \frac{1}{2}\mathrm{tr}\left(I - \frac{\Sigma^{-1}\mu\mu^\top}{1 + \mu^\top\Sigma^{-1}\mu}\right) \\
&\quad + \frac{1}{2}\log\left[\frac{\det\Sigma(1 + \mu^\top\Sigma^{-1}\mu)}{\det\Sigma}\right] - \frac{H}{2} \\
&= \frac{1}{2}\mu^\top\Sigma^{-1}\mu\left(1 - \frac{\mu^\top\Sigma^{-1}\mu}{1 + \mu^\top\Sigma^{-1}\mu}\right) \\
&\quad + \frac{H}{2} - \frac{1}{2}\frac{\mathrm{tr}(\Sigma^{-1}\mu\mu^\top)}{1 + \mu^\top\Sigma^{-1}\mu} \\
&\quad + \frac{1}{2}\log(1 + \mu^\top\Sigma^{-1}\mu) - \frac{H}{2} \\
&= \frac{1}{2}\frac{\mu^\top\Sigma^{-1}\mu}{1 + \mu^\top\Sigma^{-1}\mu} \\
&\quad - \frac{1}{2}\frac{\mu^\top\Sigma^{-1}\mu}{1 + \mu^\top\Sigma^{-1}\mu} \\
&\quad + \frac{1}{2}\log(1 + \mu^\top\Sigma^{-1}\mu) \\
&= \frac{1}{2}\log(1 + \mu^\top\Sigma^{-1}\mu).
\end{aligned}
$$

## D  NEGATIVE LOG-LIKELIHOOD ON UCI DATASETS

Here we list the test NLL scores of all models tested on all UCI datasets (Table 2). Given a test set with $N_*$ data points, if $y_i$ is the true output of the $i$th point and $\mu_i$ and $\sigma_i^2$ are a model's predictive mean and variance (including the noise), then the RMSE and NLL of the model are given by

$$
\mathrm{RMSE} = \sqrt{\frac{1}{N_*}\sum_{i=1}^{N_*}(y_i - \mu_i)^2}, \qquad \mathrm{NLL} = \frac{1}{N_*}\sum_{i=1}^{N_*}\left[\frac{(y_i - \mu_i)^2}{2\sigma_i^2} + \frac{1}{2}\log\sigma_i^2\right] + \frac{1}{2}\log(2\pi),
$$

where we note that the NLL is normalized by the number of data points. DEKL achieves lower NLL than does DE on some of the datasets, suggesting that DEKL is a method that is sometimes worth trying if DE fails to give satisfactory performance. We note, however, that DEKL achieves lower RMSE (Table 1) than does DE more often than it does lower NLL; as RMSE is perhaps the more relevant metric to measure mean predictive performance, we conclude that DEKL is certainly a method worth keeping in the practitioner's toolbox.

Table 2: Test-NLL (mean and std. dev. across 20 train-test splits) on UCI datasets, using deep ensembles (DE), deep kernel learning (DKL) and deep ensemble kernel learning (DEKL). Suffices -I and -O indicate using $KL_I$ (Eq. 11) and $KL_O$ (Eq. 10), respectively.

| Dataset | DE | DKL-I | DKL-O | DEKL-I | DEKL-O |
|---|---|---|---|---|---|
| Boston housing | $2.51 \pm 0.23$ | $2.59 \pm 0.19$ | $2.66 \pm 0.40$ | $2.54 \pm 0.21$ | $2.54 \pm 0.21$ |
| Concrete | $3.08 \pm 0.28$ | $3.01 \pm 0.14$ | $2.99 \pm 0.13$ | $2.94 \pm 0.11$ | $2.95 \pm 0.12$ |
| Energy | $1.66 \pm 1.38$ | $1.13 \pm 0.55$ | $1.24 \pm 0.66$ | $0.68 \pm 0.09$ | $0.67 \pm 0.07$ |
| Kin8nm | $-1.26 \pm 0.02$ | $-1.13 \pm 0.04$ | $-1.12 \pm 0.05$ | $-1.16 \pm 0.02$ | $-1.16 \pm 0.02$ |
| Naval Propulsion | $-6.69 \pm 0.13$ | $-5.65 \pm 4.24$ | $-6.28 \pm 1.01$ | $-6.52 \pm 0.96$ | $-6.78 \pm 0.60$ |
| Power plant | $2.77 \pm 0.05$ | $2.76 \pm 0.07$ | $2.77 \pm 0.06$ | $2.78 \pm 0.06$ | $2.78 \pm 0.05$ |
| Protein | $2.75 \pm 0.05$ | $2.82 \pm 0.03$ | $2.79 \pm 0.02$ | $2.81 \pm 0.03$ | $2.79 \pm 0.01$ |
| Wine | $0.96 \pm 0.10$ | $1.03 \pm 0.23$ | $1.01 \pm 0.09$ | $0.96 \pm 0.07$ | $0.97 \pm 0.07$ |
| Yacht | $0.21 \pm 0.17$ | $2.06 \pm 1.33$ | $1.78 \pm 0.81$ | $1.07 \pm 0.06$ | $1.08 \pm 0.07$ |

## E    REGRESSION RESULTS FOR VARIOUS NUMBERS OF LEARNERS

On the UCI datasets, we evaluate DE and DEKL (both with and without optimal prior covariance) for a varying number of learners to see its effect on test performance (Figs. 4-5). DEKL often maintains superior performance over DE in terms of RMSE as the number of learners is increased, although any trends in the performance gap itself are unclear. We believe an in-depth study on more complex problems and a much greater number of learners is required to see clearer trends, which we leave for future work.

## F    DIVERSITY OF LEARNERS

Here we present the results of the experiment described in Sec. 3.3 on all UCI datasets (Figs. 6-8). Overall, DEKL tends to have more diversity than DE, sometimes by a very large margin; for example, on Kin8nm, the DE learners are almost perfectly correlated while the DEKL learners are almost orthogonal. However, the relationship between diversity and predictive performance is more nebulous. On Boston housing, Concrete, Power plant, and Protein, DEKL has both more diversity and lower RMSE (Table 1) than DE. However, on Energy and Yacht, both DE and DEKL have almost perfect correlation among their respective learners, but DEKL achieves lower RMSE; on the other hand, on Kin8nm, Naval propulsion, and Wine, DEKL has greater diversity but not lower RMSE. Understanding when diversity is beneficial for predictive performance is a topic we leave for future work.

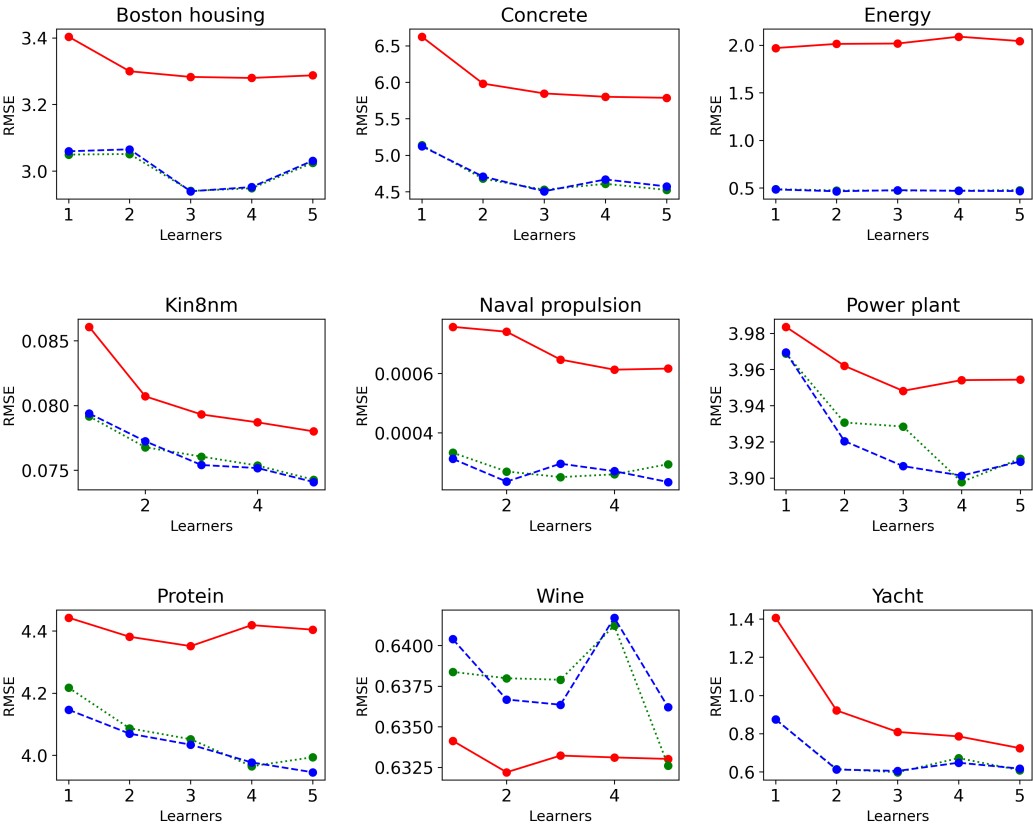

Figure 4: Mean test RMSE (across all train-test splits) of the DE (red, solid), DEKL-I (green, dotted), and DEKL-O (blue, dashed) models on nine UCI datasets as a function of the number of learners.

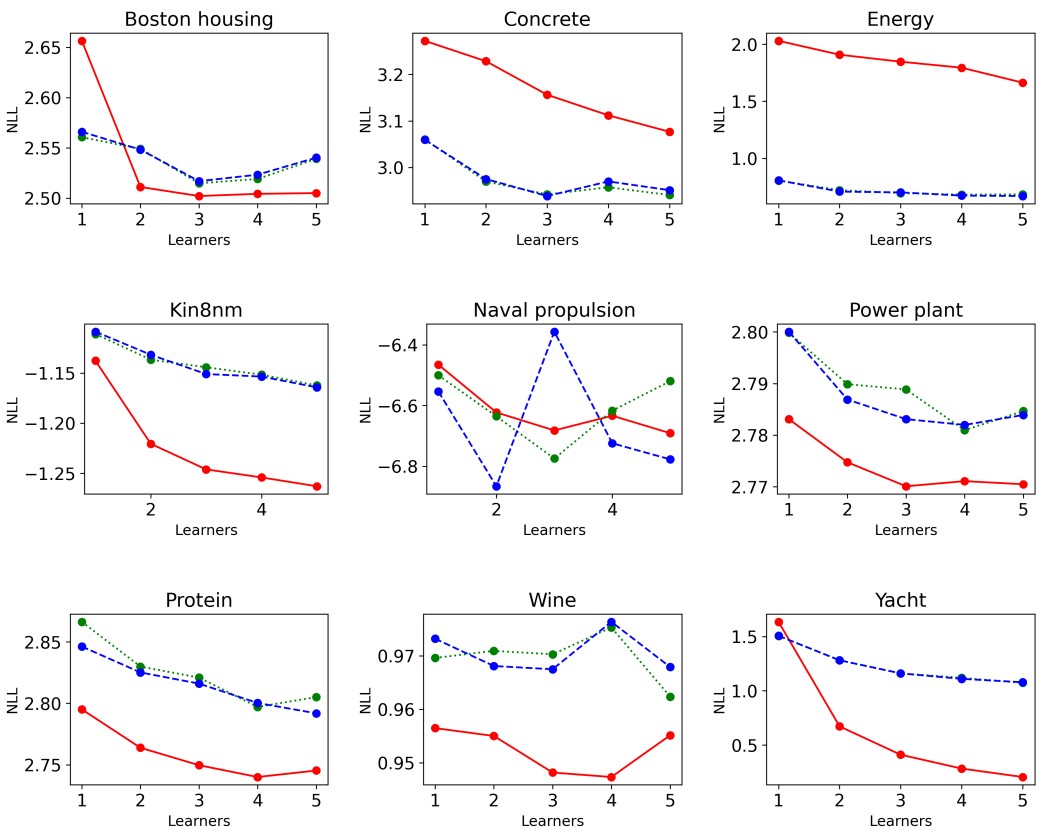

Figure 5: Mean test NLL (across all train-test splits) of the DE (red, solid), DEKL-I (green, dotted), and DEKL-O (blue, dashed) models on nine UCI datasets as a function of the number of learners.

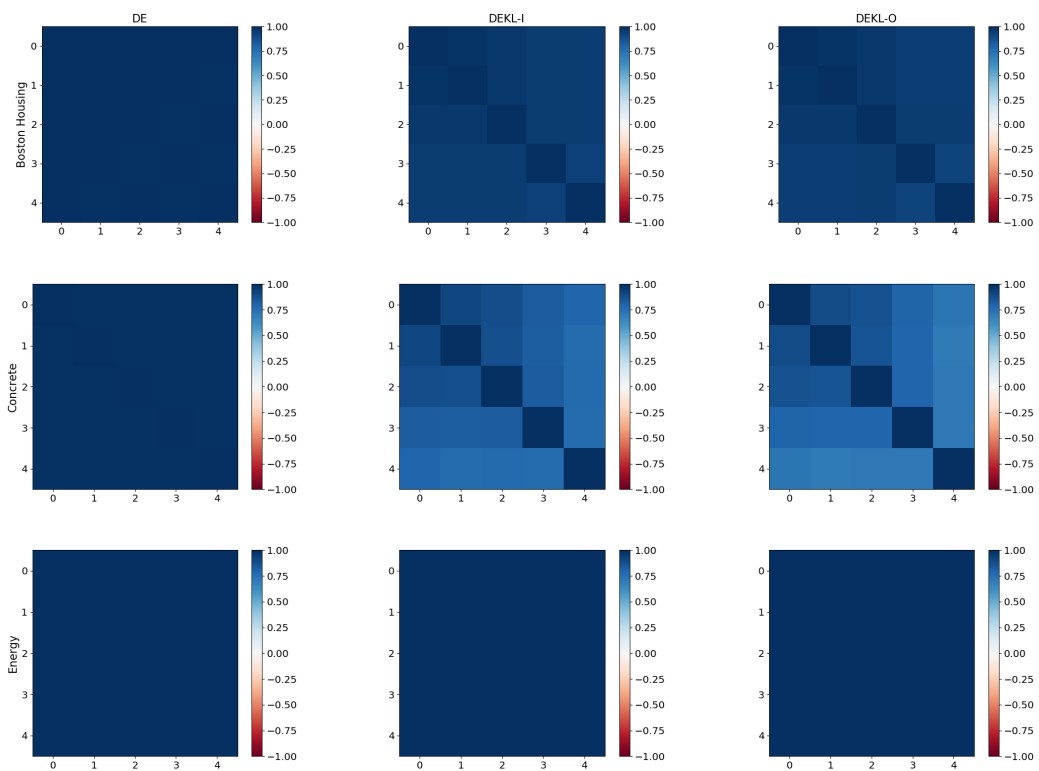

Figure 6: Correlation matrices of learners averaged over all train-test splits on the Boston housing, Concrete, and Energy datasets using deep ensembles (DE) and deep ensemble kernel learning (DEKL). Suffices -I and -O indicate using $\mathrm{KL}_I$ (Eq. 11) and $\mathrm{KL}_O$ (Eq. 10), respectively.

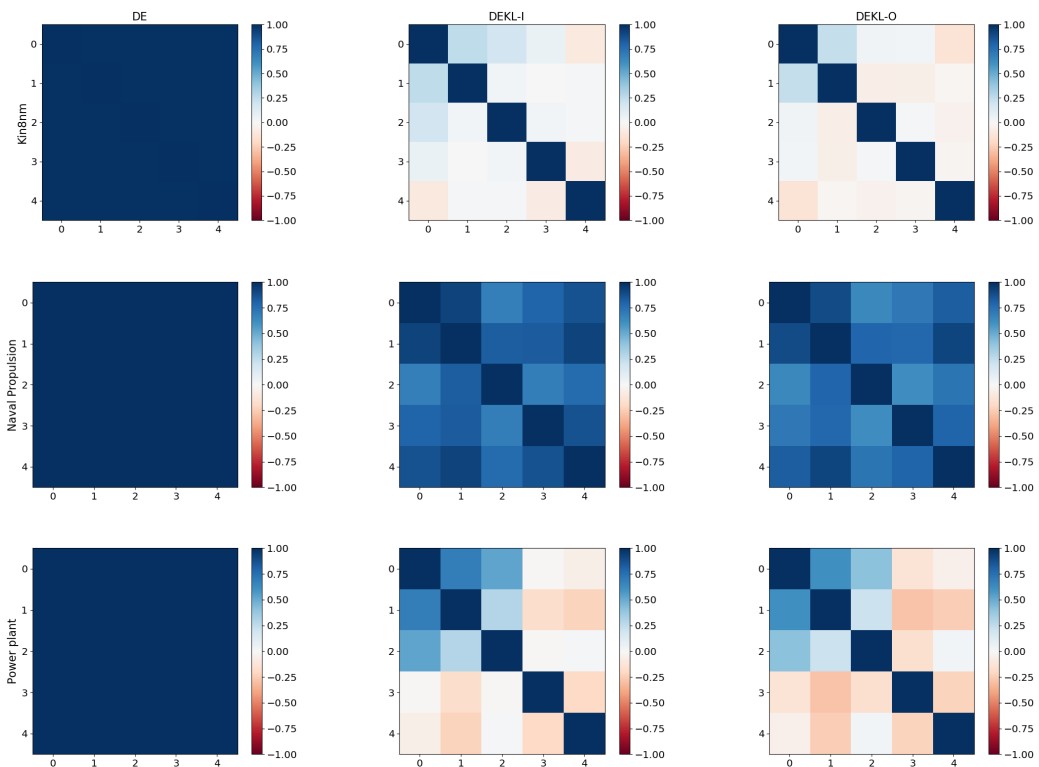

Figure 7: Correlation matrices of learners averaged over all train-test splits on the Kin8nm, Naval propulsion, and Power plant datasets using deep ensembles (DE) and deep ensemble kernel learning (DEKL). Suffices -I and -O indicate using $\mathrm{KL}_I$ (Eq. 11) and $\mathrm{KL}_O$ (Eq. 10), respectively.

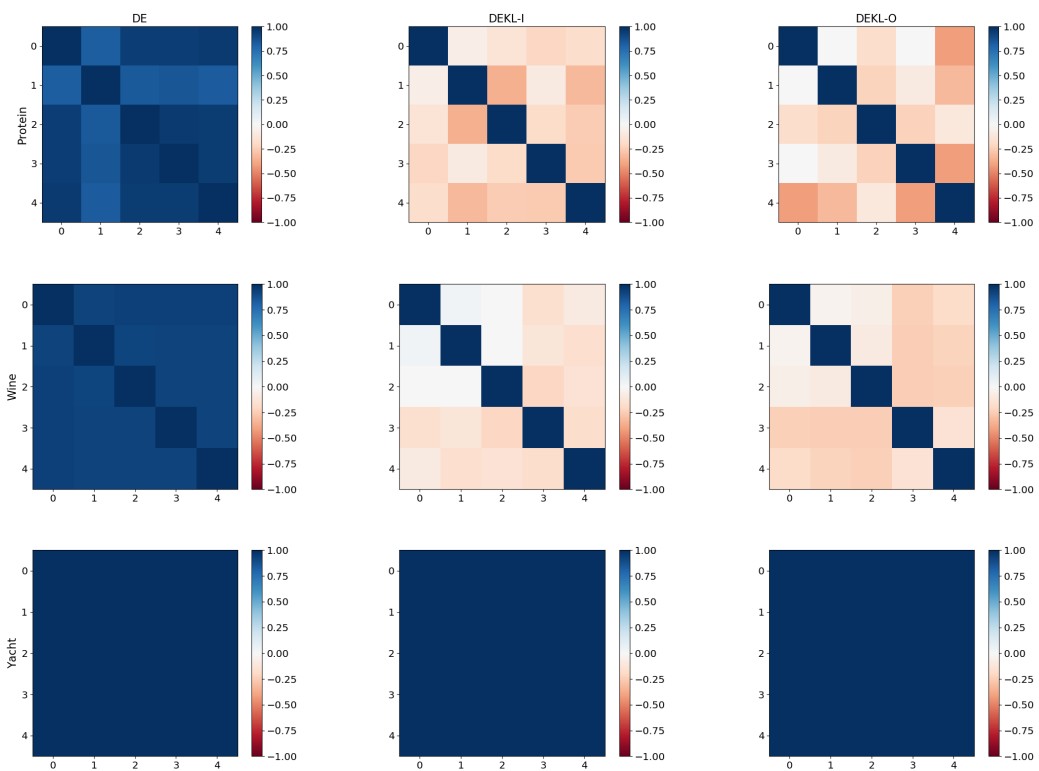

Figure 8: Correlation matrices of learners averaged over all train-test splits on the Protein, Wine, and Yacht datasets using deep ensembles (DE) and deep ensemble kernel learning (DEKL). Suffices -I and -O indicate using $\mathrm{KL}_I$ (Eq. 11) and $\mathrm{KL}_O$ (Eq. 10), respectively.

