# OpenReview forum: "Deep Ensemble Kernel Learning"
_ICLR.cc/2021/Conference — Reject_

### Official Review · AnonReviewer2 · 2020-10-26
**interesting, well-presented method; could improve literature review and make experimental section more thorough**

**Rating:** 6
**Confidence:** 4

**Review:**

This paper proposes a special case of deep kernel learning (DKL) using a linear base kernel, where the inputs to the kernel are the outputs of neural nets with identical architectures (which could be thought of as learners that get ensembled); the resulting approach is called deep ensemble kernel learning (DEKL). The paper provides a universal kernel approximating result for DEKL (Theorem 1), explains how to train DEKL efficiently for large datasets using variational inference, and demonstrates that DEKL can outperform deep ensembles and DKL on various datasets.

Overall, this paper is well-written and easy to follow, and appears to combine some standard ensembling ideas with DKL. I think the paper would benefit from more extensive literature review and a more thorough experimental results section. See the "weaknesses" listed below for details. In terms of significance, I find this paper to constitute more of an incremental advance.

Strengths:
- very well-presented, intuitive method
- theory and experiments are easy to follow

Weaknesses:
- Theorem 1 seems straightforward and rather unsurprising given the Universal Approximation Theorem; moreover, approximating the kernel to be of the form phi(x)^T phi(x) (so taking V to be the identity matrix) is already known to be able to approximate a variety of kernels with arbitrary accuracy when phi is chosen with appropriate randomness (see Rahimi and Recht’s "Random Features for Large-Scale Kernel Machines" (2007) and also the follow-up paper "Weighted Sums of Random Kitchen Sinks: Replacing minimization with randomization in learning" (2008)) -- perhaps it would be helpful to discuss a bit how your approach relates to these existing methods (there seem to be some similar ideas being used); naturally, a related question I have is how much performance degrades if V is just taken to be identity rather than using the choice given in equation (7)
- for the experiments, providing some basic characteristics of the UCI datasets used would be helpful (e.g., dataset size and dimensionality), to get a sense of how high-dimensional the datasets are
- I didn't really get intuition for when/why DEKL should outperform DKL (especially in terms of dataset characteristics), and am left wondering how much simpler methods would fare on the UCI datasets (e.g., linear regression, random forest regression, or other classical methods); for example, is DKL optimizing over too rich a model class or something?

Typo right after equation (2): $\varphi$ is missing a close parenthesis

---

> ### Author Response · Authors · 2020-11-12
> **Response to Reviewer 2**
>
> We thank the reviewer for their helpful comments and suggestions. Below we respond to each point raised, one-by-one.
>
> 1) Need more thorough literature review and more extensive experimental results section.
>
> We thank the reviewer for pointing out some relevant works in the literature (see response to Comment 2); we have included these in our revised manuscript. We have also included additional experimental results (see Sec. 3.3)-- in particular, results showing that our proposed method can promote more diversity among its learners than the baseline deep ensemble (DE) method.
>
> 2) How does DEKL relate to the methods of Rahimi and Recht’s "Random Features for Large-Scale Kernel Machines" (2007) and also the follow-up paper "Weighted Sums of Random Kitchen Sinks: Replacing minimization with randomization in learning" (2008))?
>
> We are grateful to the reviewer for pointing out our blunder of not discussing these important works. We are of course aware of these works, and it was our mistake to overlook them. In our revised manuscript, we contrast our method to that of Rahimi and Recht multiple times in Sec. 2. First, our universality theorem holds for any continuous kernel, not just stationary ones. Second, our approach differs in how we apply the universality theorem in DEKL; instead of using random features, we train them based on data, and we do so in a Bayesian framework; we suspect that optimizing the features may reduce the number of features required, compared to the random features approach.
>
> The reviewer also points out that most kernels can be expressed in the form K(x, x') = phi(x)^T phi(x') and asks why we can't set V = I in our method. The reason is that our learners in DEKL are not arbitrarily complex and thus cannot learn an orthonormal basis; that is, the concatenated learners in DEKL are not free to express any feature map phi, and the non-diagonal matrix V is needed for additional flexibility. We address this in more detail in the last paragraph before Sec 2.1 and in Appendix B in our revised manuscript.
>
> 3) How much does performance degrade when V is set to I?
>
> We have tested this in our experiments (see DEKL-I vs. DEKL-O in Table 1 for example; also in Fig. 2, where we assume V = I when the prior covariance is not optimized). On the UCI datasets, performance does not degrade significantly (see our response to comment 2 of Reviewer 4).
>
> 4) It would be useful to list the size and dimension of each UCI dataset.
>
> We agree, and we have done so in our revised manuscript (see Table 1).
>
> 5) What is the intuition for when/why DEKL can outperform DKL, in terms of dataset characteristics? Could DKL be too expressive? How would simpler methods such as linear regression or random forests fare?
>
> Understanding the performance gap between DKL and DEKL is a very interesting question that we plan to explore in more depth in future work. Perhaps it is because DEKL is less flexible due to its partitioned feature network, as the reviewer surmises. However, we do not think that even simpler models such as the classical methods the reviewer lists will work as well as DEKL. Salimbeni et al (2017) performed linear regression on the UCI datasets as one baseline, using the same experimental setup as ours (although they only report log-likelihood), and they found that GP models outperform linear regression on most of the datasets. Moreover, in Appendix E in our revised manuscript, we show the results for various numbers of learners and we see, for instance, that using only one learner (i.e., a narrow DKL model) does not perform as well as five learners. This suggests that some complexity in our models is important on the UCI dataset tasks.
>
> 6) Typo right after Eq. 2; phi is missing a close parenthesis.
>
> We have fixed this in the revised manuscript.

---

### Official Review · AnonReviewer4 · 2020-10-28
**An interesting contribution well-founded from both mathematical background and computational experiments**

**Rating:** 8
**Confidence:** 4

**Review:**

The authors introduce a deep ensemble kernel learning approach as a linear-based learning combination, from a deep learning scheme, to approximate kernel functions under a Bayesian (GP) framework. Namely, a universal kernel approximation strategy is proposed from eigen-based decomposition and deep learning-based function composition. Then, a variational inference strategy is used to solve the optimization from kernel-based mappings. Two regularization strategies are studied: optimal prior covariance and isotropic covariance. Results demonstrate the benefits of the proposal.

Two comments:
Could you include the GP and sparse GP results to compare the predictions? Your approach can be scalable from the stochastic variational variance; however, the studied databases are small in terms of the number of samples so that the well-known GP algorithms could be compared.
In most of the cases, isotropic and optimal covariance-based methods seem to achieve the same results; why?

---

> ### Author Response · Authors · 2020-11-12
> **Response to Reviewer 4**
>
> We thank the reviewer for their helpful comments and suggestions. Below we respond to each point raised, one-by-one.
>
> 1) Include GP and sparse GP results to compare predictions.
>
> Salimbeni et al (2017) (see references in our manuscript) have already tested GP methods on the UCI datasets, using the same experimental setup as we did, and in our revised manuscript, we have added a reference to their results and compare them with ours. We find that DEKL either outperforms their best GP model or attains a comparable score (within standard deviation) on all UCI datasets considered.
>
> 2) Why do the identity prior covariance (KL_I) and optimal prior covariance (KL_O) regularizers lead to comparable performance?
>
> This is a very interesting question that we plan to explore in more depth in future work, but in our revised manuscript, we have added a brief discussion speculating on the reason (see Sec. 3.2 and the end of 3.3): Optimizing V is equivalent to finding an optimal basis on the span of the learners {phi_i}. However, if the learners are sufficiently flexible, then we may assume without loss of generality that V = I. We therefore suspect that on the UCI datasets, the network architecture we chose for the learners is sufficiently flexible to find an optimal basis on their span. Indeed, in Sec. 3.3 we have included new results showing that the learners are often able to orthogonalize during training. An alternative explanation is that the UCI datasets are such that the difference between the KL_I and KL_O regularizers is insignificant compared to the contribution of the expected log-likelihood term in the loss function. As contrast, note the difference in the effects of the KL_I and KL_O regularizers on the synthetic cubic dataset, where the stronger KL_I regularizer is more beneficial.

---

### Official Review · AnonReviewer3 · 2020-10-28
**see review**

**Rating:** 5
**Confidence:** 3

**Review:**

This paper proposes a variant of the Deep Kernel Learning model (DKL) [1] where multiple independent networks are trained for the features instead of a single network. In addition, the paper proposes to use a linear kernel as a base kernel which allows for universal approximation of any arbitrary kernel, as well as allowing for exact inference of the kernel hyperparameters. The use of stochastic variational inference is proposed for inferring the neural networks weights. The model is compared against Deep Ensemble (DE) and DKL on a synthetic dataset and the UCI dataset.

The paper is well written, easy to follow and is technically correct. The main weaknesses of the paper is that by using a linear kernel as the base kernel, it is not clear how the proposed model is different from a Bayesian Neural Network with an extra linear layer. Furthermore, the experiments do not necessarily showcase the strengths of the proposed model, namely, the universal approximation property and the regularization that an ensemble provides. I think the paper could increase its impact by effectively implementing distributed training as well as improving the experiments.

[1] Andrew Gordon Wilson, Zhiting Hu, Ruslan Salakhutdinov, and Eric P Xing. Deep kernel learning.
In Artificial Intelligence and Statistics, pp. 370–378, 2016b.

---

> ### Author Response · Authors · 2020-11-12
> **Response to Reviewer 3**
>
> We thank the reviewer for their helpful comments and suggestions. Below we respond to each point raised, one-by-one.
>
> 1) By using a linear kernel as the base kernel, it is not clear how DEKL is different from a Bayesian neural network (BNN) with an extra linear layer.
>
> We would like to clarify that our DEKL model is quite different from a BNN. Unlike in a BNN, we only perform Bayesian inference on the last layer of our model. In contrast, we treat the parameters of the feature network as prior hyperparameters.
>
> 2) Experiments do not showcase the strengths of DEKL-- Universal kernel approximation and regularization through ensembling.
>
> In our revised manuscript, we have included new experimental results illustrating how DEKL promotes more diversity among its learners than does the deep ensemble (DE) baseline where all learners are independent (see Sec. 3.3). This provides some insight into the regularization effect of DEKL. We did not empirically study convergence in our universality theorem because we felt the proof was sufficient, as other reviewers have pointed out.
>
> 3) Paper could increase impact by implementing distributed training and improving experiments.
>
> The reviewer is of course right that a distributed implementation is the natural next step, as it is the ultimate upshot of our method. However, distributed training is likely beyond the scope of this paper, but we point out that it is fairly easy to see that DEKL is more easily parallelizable than DKL, suggesting that a distributed implementation will likely give us the results we expect. Regarding additional experiments, we have included new experiments on the diversity of learners (see response to Comment 2).

---

### Official Review · AnonReviewer1 · 2020-10-30
**DKL with linear kernel**

**Rating:** 3
**Confidence:** 5

**Review:**


In this paper, an extension to deep kernel learning is proposed. A linear kernel is used as the base kernel, which enables exact optimization of the kernel hyperparameters. There is a universal approximator theorem stating that the deep neural network with linear kernel could approximate any kernel function, which is quite obvious from the perspective of random Fourier features as well. Also, multiple neural networks are used to produce features and the features are concatenated.  I am not sure why the word ensemble is used, but it is really a concatenation of features instead of ensembling predictions. Besides the exact inference, standard variational inference is also proposed for the linear base kernel. Experiments are conducted on synthetic data and UCI datasets, with comparison to DKL with linear kernel and deep ensembles.
In general, I could not find very interesting contributions from the paper. The framework follows closely from DKL with a linear kernel. But I have a question about the proof of the universal approximator theorem, to approximate the different eigenfunctions, I would assume the hidden layer of the neural network to be at least of O(B) where B is the number of eigenfunctions to be approximated. So potentially, the neural network needs to be very wide which might not be practical. Also, I am confused why the “concatenation” of features is referred to as “ensemble” in the paper. It is super confusing to me unless I am not understanding how the features are used jointly. Also, the authors mention that “a learner with M output is simply a concatenation of M single-output learners, suggesting that multi-output learner may help to further reduce the number of H of required learners.” With the same number of nodes in the hidden layers, I don’t think a multi-output learner is equivalent to M different single-output learners, therefore the argument made here might not be valid.
In terms of experiments, it is weird that for DKL linear kernel is used instead of the original spectral mixture kernel with random Fourier features. It makes the most sense to compare the linear kernel with something like RBF or spectral mixture kernel. Also, UCI regression tasks seem to be rather easy and do not necessarily need a deep neural network to do feature engineering. It would be necessary to conduct experiments on more complicated tasks such as images to validate the effectiveness of the proposed linear kernel approach.

------------After author's response----------------

My major concern is about the connection between the universal approximation theorem and the proposed architecture. In the paper, the authors mentioned that " the following universal approximation theorem for DKL implies that this effect can be compensated by adding parallel learners". However, the fact that a multi-output learner is not equivalent to M different single-output learners makes it hard to justify the proposed architecture theoretically from the universal approximation theorem.
I think this is something that is crucial to be justified, otherwise, the theory does not really match with the proposed method.

---

> ### Author Response · Authors · 2020-11-12
> **Response to reviewer 1**
>
> We thank the reviewer for their helpful comments and suggestions. Below we respond to each point raised, one-by-one.
>
> 1) The work follows too closely to DKL with a linear base kernel.
>
> We see now that our manuscript lacked clarity in its discussion around the linear base kernel, and we thank the reviewer for pointing this out. In our revised manuscript, we show that if we replace the linear base kernel with any bounded or half-bounded kernel, such as the RBF, then the resulting deep kernel is no longer a universal kernel approximator (see Remark 2 and Appendix B). This suggests that the linear base kernel is quite special, and if the goal of DKL/DEKL is to be sufficiently flexible to parameterize a large class of kernels, then the linear base kernel should be preferred over the RBF for example.
>
> Moreover, our proposed method goes beyond DKL with a linear base kernel by partitioning the feature network into an "ensemble" of learners, thus allowing for easier model parallelism.
>
> 2) The universal kernel approximation theorem is obvious from the perspective of random Fourier features.
>
> We thank the reviewer for bringing to our attention the work of Rahimi and Recht; it was our blunder not to include this work in the discussion. We note that in contrast to the random Fourier features method, our universality theorem holds even for non-stationary kernels. Moreover, in practice, we optimize our features, which could help to reduce the total number of features needed compared to random features. We have modified our manuscript to include this discussion.
>
> 3) Not sure why the word "ensemble" is used. It is a concatenation, not an ensemble.
>
> We see how this can be a point of confusion, and we have tried to clarify this in our revised manuscript. We use the word "ensemble" because we can think of DEKL as an extension of the deep ensemble (DE) method of Lakshminarayanan et al (2017); the main difference is that in our proposed method, the learners are not entirely independent because they feed into a common final layer. The word "ensemble" thus provokes a comparison to the DE method, and we hypothesize that our method is superior because the final layer can help to promote diversity among the learners, thus boosting robustness. We have added more experimental results in our revised manuscript, where we show that our method indeed achieves greater diversity among its learners than does DE (see Sec. 3.3).
>
> 4) The feature network may need to be very wide, which may not be practical.
>
> This is certainly true for DKL, but not for our proposed method DEKL. In our method, we partition the feature network into an "ensemble" of learners, each of which may be narrow. Although the total number of learners in the ensemble may need to be large, we could imagine distributing the learners across ranks in a distributed computing environment to leverage their disjointedness.
>
> 5) An M-output learner is not equivalent to a concatenation of M single-output learners.
>
> We agree this is true in general. We would like to clarify that in our manuscript, we state that an M-output learner is the concatenation of M single-output learners only when each learner is a single-output affine map (followed by an activation function)-- i.e., no hidden layers. Although this is trivial, it suggests that in the case of learners with deeper architectures, using multi-output learners could still help to reduce the total number of required learners at least to some extent.
>
> 6) It would make more sense to compare to DKL with RBF or spectral mixture kernel.
>
> The reviewer is right to point out that the most popular DKL models use the RBF or spectral mixture kernels as the base kernel. As already discussed in our response to Comment 1, however, at least the RBF kernel may not be the ideal base kernel if we want our model to be a universal kernel approximator.
>
> Moreover, as already mentioned, the purpose of moving from the DKL model to our proposed DEKL model is to reduce computational complexity by partitioning the feature network. However, we want to ensure that we do not sacrifice predictive performance in the process. To test this, we need to compare DEKL to the DKL baseline using the same base kernel, namely the linear kernel.
>
> 7) UCI datasets are too easy for neural networks. Consider more complicated tasks such as images.
>
> Certainly additional example applications would provide more evidence for the utility of our proposed method. However, due to limited space, we stuck to the UCI datasets, which have been the most popular benchmark tasks for Bayesian deep learning. These datasets are diverse, with their sizes raising from 100's to 10000's. We certainly plan to investigate more complex tasks in the future, especially once we have a distributed implementation of DEKL.

---

### Decision · Program_Chairs · 2021-01-07
**Final Decision**

**Decision:**

Reject

**Comment:**

The paper presents a DKL variant with a linear kernel. Representations from several networks is combined through concatenation, making it not quite an ensemble. It's shown that the model is a universal kernel approximator. Experiments are conducted on a large number of UCI datasets.

Following the discussions, the paper still has the following shortcomings:
- some lack of clarity in the presentation (for instance, explaining the equivalence between a multi-output learner and M different single-output learners)
- lack of experiments on data where deep learning is typically used (images); the UCI datasets have structured data and other ensembles like XGBoost may outperform the baselines presented in this paper
- difference in performance between DKL and DEKL, especially since DKL benefits from a larger model space, theoretically. maybe DEKL has better sample complexity, but does this advantage hold in the case of the large datasets that deep learning is used for?